# Tele-medicine controlled hospital at home is associated with better outcomes than hospital stay

Noa Zychlinski[1], Ronen Fluss[2], Yair Goldberg[1], Daniel Zubli[3], Galia Barkai[3], Eyal Zimlichman[4], Gad Segal [3,5,6] *

1 Faculty of Data and Decision Sciences, Technion–Israel Institute of Technology, Haifa, Israel,
2 Biostatistics and Biomathematics Unit, Gertner Institute of Epidemiology and Health Policy Research, Chaim Sheba Medical Center, Tel Hashomer, Israel, 3 Sheba Beyond Virtual Hospital, Chaim Sheba Medical Center, Ramat Gan, Israel, 4 Management Wing, Chaim Sheba Medical Center, Ramat Gan, Israel, 5 Education Authority, Chaim Sheba Medical Center, Ramat Gan, Israel, 6 Faculty of Healthcare and Medicine, Tel Aviv University, Tel-Aviv, Israel

* Gad.segal@sheba.health.gov.il

**Data Availability Statement:** All de-identified data will become available upon request from the Principal Investigator (Prof. Gad Segal, MD). This is a mandatory, legal term on behalf of our

## Abstract

### Background

Hospital-at-home (HAH) is increasingly becoming an alternative for in-hospital stay in selected clinical scenarios. Nevertheless, there is still a question whether HAH could be a viable option for acutely ill patients, otherwise hospitalized in departments of general-internal medicine.

### Methods

This was a retrospective matched study, conducted at a telemedicine controlled HAH department, being part of a tertiary medical center. The objective was to compare clinical outcomes of acutely ill patients (both COVID-19 and non-COVID) admitted to either in-hospital or HAH. Non-COVID patients had one of three acute infectious diseases: urinary tract infections (UTI, either lower or upper), pneumonia, or cellulitis.

### Results

The analysis involved 159 HAH patients (64 COVID-19 and 95 non-COVID) who were compared to a matched sample of in-hospital patients (192 COVID-19 and 285 non-COVID). The median length-of-hospital stay (LOS) was 2 days shorter in the HAH for both COVID-19 patients (95% CI: 1–3; p = 0.008) and non-COVID patients (95% CI; 1–3; p < 0.001). The readmission rates within 30 days were not significantly different for both COVID-19 patients (Odds Ratio (OR) = 1; 95% CI: 0.49–2.04; p = 1) and non-COVID patients (OR = 0.7; 95% CI; 0.39–1.28; p = 0.25). The differences remained insignificant within one year. The risk of death within 30 days was significantly lower in the HAH group for COVID-19 patients (OR = 0.34; 95% CI: 0.11–0.86; p = 0.018) and non-COVID patients (OR = 0.38; 95% CI: 0.14–0.9; p = 0.019). For one year survival period, the differences were significant for COVID-19

institutional ethics committee. Upon request, data will go through a double check of anonymization and then sent direct to whoever requested the data. Aside from the Principal Investigator, a non-author contact that can be addressed is our local IRB at the following email address: Helsinki@sheba.health.gov.il. The aforementioned restrictions are warranted since the patients' data could potentially include sensitive patients' information.

**Funding:** The Research was supported in part by an Israel Science Foundation [Grant 277/21] and the Israel National Institute for Health Policy Research [Grant 2021/160/R]. Guarantor: No guarantees were given regarding this study.

**Competing interests:** The authors have declared that no competing interests exist.

patients (OR = 0.5; 95% CI: 0.31–0.9; p = 0.044) and insignificant for non-COVID patients (OR = 0.63; 95% CI: 0.4–1; p = 0.052).

## Conclusions

Care for acutely ill patients in the setting of telemedicine-based hospital at home has the potential to reduce hospitalization length without increasing readmission risk and to reduce both 30 days and one-year mortality rates.

## Introduction

### Hospital-at-home (HAH) services worldwide

Globally, healthcare systems in general and their hospitalization arms in particular, are experiencing hardships in terms of infrastructure, resources, and lower availability of skilled healthcare professionals. These hardships were worsened, as stated by the World Economic Forum, by the unprecedented disruptions caused by the COVID-19 pandemic [1]. As a result, it was recently published, in a 2023 survey, that 46% of adults worldwide encounter limited access to treatment and prolonged waiting times to reach affordable health resources with lack of staff being the biggest challenge [2]. These challenges are being answered by social, financial, and healthcare organizations, with innovative approaches and solutions being advocated. One such approach was recently introduced by a global consulting firm, presenting the concept of hospitals without walls [3]. This wide-span concept of health without boundaries, includes the adoption of advanced high-technology in the service of telemedicine, serving as an enabler for making the HAH services the safest and most effective as can be attained. Recent years brought success in this realm, mainly with regard to COVID-19 patients [4–6].

### Telemedicine-controlled hospital-at-home services

A predominantly important factor contributing to the prognosis of patients during hospitalization in an internal medicine department, is the experience of their attending, senior physicians. These practitioners are becoming less available and practically inexistent in some peripheral areas. One way of coping with this problem would depend on the ability of experienced, senior internal-medicine specialists to diagnose and treat their patients from a distance, upscaling their influence on population health. Recent advancements in telemedicine, another consequence of the COVID-19 pandemic, have paved the way for sophisticated remote medical services, introducing home hospitalization as a viable alternative to traditional on-site care. Recently published studies' results, relating to the advantages of tele-monitoring and miniaturized technologies, either in the HAH settings or post-hospital-discharges plans, enhance our ability of rely on tele-medicine services in terms of patients' safety, risks of re-admission and other, in-hospital related side events [7–9].

During the year 2020, Sheba Beyond was established as an integral part of the Sheba Medical Center, encompassing all tele-health services in this tertiary hospital. By enabling remote physical examinations, monitoring, and online rehabilitation programs, Sheba Beyond aims to make high-quality medical expertise accessible to broader audiences. This aligns with the growing expectation that remote hospitalization will become a widely available service among major hospital networks, across many specialties. During the past several years, the unique HAH service at Sheba Beyond served not only as a clinical service but also as a validation

laboratory for essential, telemedicine technologies and methodologies: TytoCare© technology, serving as a remote, digital stethoscope, was clinically investigated, with measurements of physicians' compliance [10], validity and inter-observers' consensus of clinical interpretations [11]. Biobeat© technology for wireless, remote monitoring of several physiologic vital signs and parameters was validated for its reliability of telemetric transmission and comparison to overhead monitors, and potential to accumulate patients' data that could foresee future patients' deterioration [12]. A six-lead, self-handled electrocardiography (ECG) device transmitting heart rhythm description and analysis was also validated and the level of consensus of agreement was tested versus a gold-standard, legacy 12-lead ECG machine [13]. Alongside these technologies, methodologies of telemedicine based HAH were also investigated, such as a clinical pivotal trial done with a specialist in internal medicine, based within an in-hospital, internal medicine department, managed patients that stayed in their elderly home [14]. The ability of safeguarding acutely ill patients in the HAH setting was also shown to be feasible in a significant portion of patients, diagnosed as suffering from an acute, infectious disease, who demonstrate laboratory evidence of myocardial damage and still, are enjoying the efficacy and safety of the HAH service [15]. The concept of assimilating a virtual medicine-based department into the structure of a conventional medical center was also recently described [16].

## Aim of the current study

Prior research has focused on the efficacy of telemedicine-based medical services to various patients' populations including remote rehabilitation across various indications including deterioration of patients suffering from chronic congestive heart failure [17], sarcopenia [18], post-stroke recovery [19], exacerbation of chronic obstructive pulmonary disease (COPD) [20], and post-acute therapy [21]. However, limited attention has been directed towards investigating remote hospitalization of patients in the setting of acute illness, regularly directed to in-hospital stay in internal medicine wards. Some of the studies who addressed acute illness HAH services, included telemedicine visits in various proportions, however none of the programs were based on physician telemedicine visits [22–24].

This study focused on the distinctive remote telemedicine-based internal medicine model. Unlike traditional on-site admissions, patients underwent admission by a remote physician, receiving a personalized treatment plan that integrates home visits and adequate medical monitoring utilizing cutting-edge technologies. The full spectrum of nursing services was performed at patients' homes, along with laboratory testing and chest x-rays as indicated by the attending physicians who delivered service via remote, telemedicine platforms. The present study investigated the efficacy and safety of this service in a retrospective comparison and analysis of both COVID-19 and non-COVID matched patient populations.

## Methods

### Study design and patients' care

This study was performed by the Sheba Medical Center, 1,900 beds, tertiary hospital, largest of its kind in Israel. This was a retrospective matched study with 159 Sheba-Beyond hospitalizations (64 for COVID-19 and 95 for non-COVID) categorized as Group HAH. They were compared to a matched sample of controls, denoted as Group C, out of 6,817 patients who were hospitalized in the internal-medicine departments of Sheba Medical Center (2,242 for COVID-19 and 4,924 for one of three acute, infectious diseases: urinary tract infections (UTI, either lower or upper), pneumonia, or cellulitis) over the years 2021–2023 inclusive. The study included patients aged 18 and older. Respiratory and hemodynamically unstable patients as well as mild COVID-19 patients were excluded from the study. All patients' data were

extracted from their electronic medical records (EMR) which serve for clinical purposes. Fig 1 details the above patient consort flow and exclusion diagram.

Ethic statement: Data was mined after approval by a local, institutional review board (approval # SMC-21-8828) and after patients' written consent was waived due to the retrospective nature of this study. Clinical data was approached during the months between August 2023 and February 2024.

Eligible patients for home hospitalization were transitioned to receive care in the comfort of their homes. The HAH team attending these patients consists of internal medicine specialists, licensed case management nurses, home visiting nurses, X-ray technicians and call center nurses. The patient receives daily a minimum of one remote physician visit, 2 nurses visit (1 at home) and an individualized treatment plan that may include imaging, blood testing and IV, and oral treatment. Medical directives, encompassing vital signs monitoring regimen and treatments, are either carried out by the patients themselves or administered by the nursing staff during scheduled home visits. A video conversation with the attending physician was conducted at least once daily, typically in the morning, with the visiting nurse present at the patient's home. Video calls were done using a designated platform for telemedicine purposes (DATOS). During these sessions, a remote physical examination was facilitated using the Tyto-Care® system. This digital platform incorporates a digital stethoscope enabling heart, lung and abdominal (peristalsis) auscultation, a digital otoscope for visualizing the tympanic membrane, a digital thermometer, and a tongue depressor for visual examination of the pharynx. The device guides patients (or their assistants) through the examination process and records data and visuals, which are then transmitted through the internet for review by the physician. Video conferences were conducted whenever a specialist consultation was deemed necessary, with one of Sheba's specialists, as regularly done in the in-hospital settings. Each daily visit was documented in the patient's electronic medical record, including orders for blood tests, oxygen

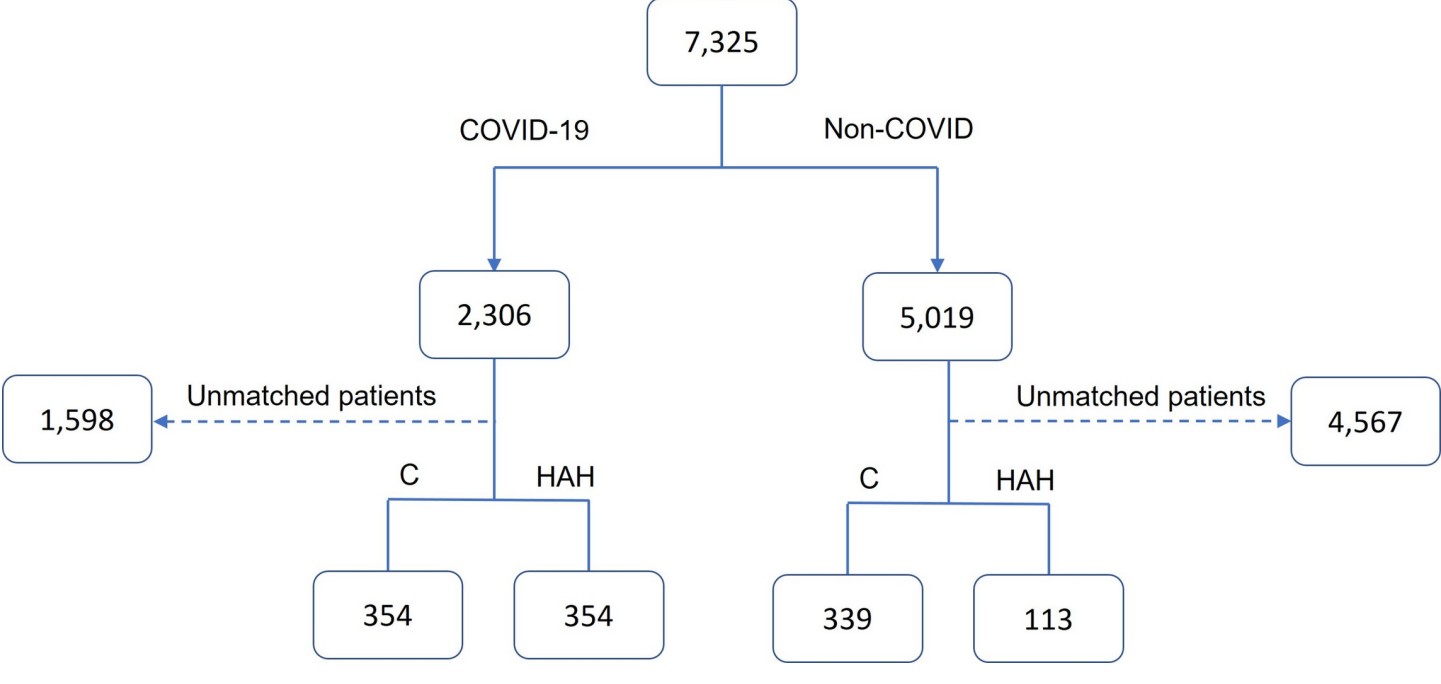

**Fig 1. CONSORT flow of patients.**

enrichment, prescribed medications, and recommendations for either hospital readmission in case of deterioration or discharge in case of improvement.

In the event of patient deterioration during home hospitalization, immediate coordination with the physician would facilitate the patient's return to the hospital's emergency department. Conversely, when the patient was ready for discharge, a discharge letter was sent, and the ongoing treatment plan was communicated to the staff via phone to ensure optimal continuity of care. The attending physician remained available for further consultation regarding the patient's care for an additional week after discharge.

## Data mining and analysis

All relevant patients' characteristics were extracted from their EMR: age, severity of disease as categorized by clinicians for COVID-19 patients, gender; chronic / background diagnoses and chronic medications. We gathered individual diagnoses to the following silos: active malignancy, past malignancy, hematologic diseases, neurologic, metabolic, cardiovascular, respiratory, autoimmune, and gastrointestinal diseases. Similarly, chronic medications were also listed and grouped, as relating to either malignant, neurologic, metabolic, cardiovascular, respiratory, autoimmune, or gastrointestinal, as well as chronic medication for ophthalmic use.

The analysis compared outcomes of COVID-19 and non-COVID patients in groups HAH and C. Clinical outcome measures included mean length of stay (LOS) in days, readmission rates within 30 days or one year from discharge and mortality rates from admission within the same time frames. In the readmission analyses, patients who died before discharge were excluded and patients who died within the follow-up period were regarded as readmissions.

We used propensity scores (PSs) to match patients from the HAH with those from C group. Four risk factors (RF) deemed relevant, up front to the study outcomes, and were therefore included in the PS for patients' matching: age, presence of active malignancy, dementia, and chronic kidney disease (CKD). For COVID-19 patients, grade of disease severity (as indicated during the period of hospitalization) was also incorporated. Additional risk factors were scrutinized individually in separate univariate logistic regressions for each one of the clinical outcomes serving as the dependent variable, while controlling for the four risk factors mentioned above. We retained those risk factors which had a p value less than 0.05 in at least one outcome. PSs were obtained, representing the estimated probability of being in the HAH group, using a logistic regression that included the RFs selected in the previous stage as predictors. Controls were matched to patients in the HAH group using the PS with a ratio of 1:3. In case of COVID-19 hospitalizations, we enforced exact matching of severity. We assessed the similarity of the resulting matched groups both graphically and by the calculating the standardized mean differences (SMD) of the RFs and PS. An absolute SMD less than 0.25 is usually regarded as a good balance [25].

A univariate analysis was used to compare the matched groups. The LOS was tested using the Wilcoxon test and the 95% CI was obtained by bootstrap resampling. The mortality rates were compared using the Fisher exact test. The readmission rates were compared using a weighted logistic regression. The weights were used to keep the balance of the matched samples after we excluded those who died before discharge. We also used Cox regression to compare time to death or readmission within one year.

## Results

Table 1 includes demographic and clinical features of all patients included in the two HAH groups and the two control groups. All the reported ratios compared the HAH group to the C group.

**Table 1. Demographic and clinical features of all study patients.**

| Feature (%) | COVID-19 | | | Non-COVID | | |
|---|---|---|---|---|---|---|
| n | Control 192 | HAH 64 | p value | Control 285 | HAH 95 | p value |
| **Patients' demographics** | | | | | | |
| Age, years (median [IQR]) | 79.25 [68.56, 87.87] | 81.00 [68.00, 86.50] | 0.809 | 80.00 [70.00, 87.00] | 83.00 [70.50, 89.00] | 0.129 |
| Male, N (%) | 84 (43.8) | 28 (43.8) | 1 | – | – | |
| **Clinical Characteristics** | | | | | | |
| Severe COVID 19 | 96 (50.0) | 32 (50.0) | 1 | – | – | |
| Active malignancy | 23 (12.0) | 7 (10.9) | 1 | 26 (9.1) | 15 (15.8) | 0.085 |
| Dementia | 24 (12.5) | 7 (10.9) | 0.828 | 42 (14.7) | 16 (16.8) | 0.623 |
| CKD | 19 (9.9) | 5 (7.8) | 0.805 | 13 (4.6) | 3 (3.2) | 0.770 |
| Hematologic | 4 (2.1) | 3 (4.7) | 0.371 | 11 (3.9) | 3 (3.2) | 1 |
| Neurologic | 34 (17.7) | 9 (14.1) | 0.567 | 61 (21.4) | 14 (14.7) | 0.182 |
| Metabolic | 92 (47.9) | 30 (46.9) | 1.000 | 155 (54.4) | 50 (52.6) | 0.812 |
| Autoimmune | 15 (7.8) | 4 (6.2) | 0.79 | – | – | |
| **Chronic medications** | | | | | | |
| Neurologic | 70 (36.5) | 21 (32.8) | 0.653 | 171 (60.0) | 50 (52.6) | 0.230 |
| Metabolic | 88 (45.8) | 28 (43.8) | 0.885 | 192 (67.4) | 63 (66.3) | 0.900 |
| Malignancy | – | – | | 69 (24.2) | 23 (24.2) | 1.000 |
| Respiratory | – | – | | 20 (7.0) | 7 (7.4) | 1.000 |
| Cardiovascular | – | – | | 193 (67.7) | 60 (63.2) | 0.452 |
| **Clinical outcomes** | | | | | | |
| Readmission (at 30 days) | 36 (23.3) | 14 (23.2) | 1 | 63 (25.7) | 18 (23.6) | 0.25 |
| Readmission (at one year) | 64 (42.2) | 26 (41.8) | 0.8 | 162 (65) | 51 (62.1) | 0.11 |
| Mortality (at 30 days) | 45 (23.4) | 6 (9.4) | 0.018 | 49 (17.2) | 7 (7.4) | 0.019 |
| Mortality (at one year) | 68 (35.4) | 14 (21.9) | 0.046 | 102 (35.8) | 25 (26.3) | 0.103 |
| LOS (median [IQR]) | 7.00 [4.00, 12.00] | 5.00 [4.00, 8.00] | 0.008 | 4.00 [2.00, 9.00] | 2.00 [2.00, 4.00] | < 0.001 |

CKD, Chronic kidney disease; LOS, Length of stay; All p values were derived using the Fisher exact test, except for age and LOS, for which the Wilcoxon rank sum test was used.–Features not included in the matching.

## Length of Stay (LOS)

The median LOS among COVID-19 patients was 5 and 7 in patients' groups HAH and C, respectively, with a statistically significant 2-day difference, and the 95% CI was 1–3 (p value = 0.008). Among non-COVID patients, the median LOS was 2 and 4 days, in groups HAH and C, respectively, with a statistically significant 2-day difference, and the 95% CI was 1–3 (p value < 0.001).

## Readmission within 30 days and one year duration

The Odds Ratio (OR) for readmission within 30 days among COVID-19 patients was 1, and the 95% CI was 0.49–2.04 (p value = 1). For non-COVID patients, the OR was 0.7 and the 95% CI was 0.39–1.28 (p value = 0.25). The OR for readmission within one year among COVID-19 patients was 1.05 and the 95% CI was 0.57–1.93 (p value = 0.8). For non-COVID patients, the OR was 0.67 and the 95% CI was 0.41–1.09 (p value = 0.11).

The Hazard Ratio (HR) for the time to readmission within one-year from discharge for COVID-19 patients was 1.06 and the 95% CI was 0.67–1.96 (p value = 0.8). For non-COVID patients, the HR was 0.81 and the 95% CI was 0.59–1.13 (p value = 0.21). Fig 2 shows the one-year Kaplan–Meier estimated survival curves for re-readmission among the COVID-19 and

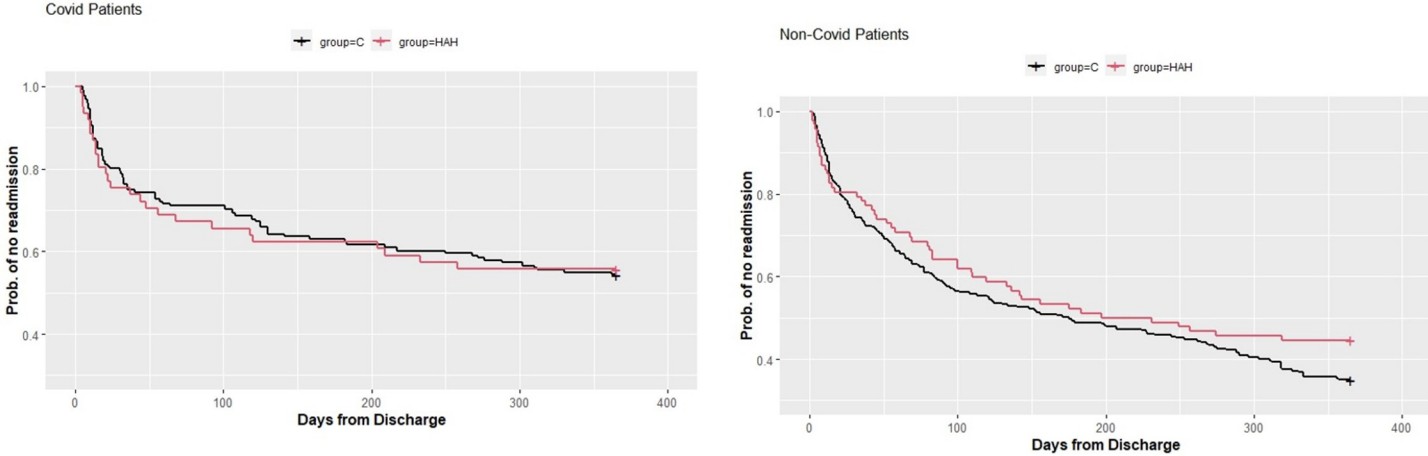

**Fig 2. Kaplan–Meier analysis for readmission probability.**

non-COVID patients: HAH versus their controls. For COVID-19 patients, the control and HAH groups are very close with no significant difference. For non-COVID, the probability of no readmission was lower, though not significantly, in the HAH group.

## Mortality within 30 days and one year duration

The OR for mortality within 30 days among COVID-19 patients was 0.34 and the 95% CI was 0.11–0.86 (p value = 0.018). Among non-COVID patients, the OR was 0.38 and the 95% CI was 0.14–0.90 (p value = 0.019). The OR for mortality within one year among COVID-19 patients was 0.51 and the 95% CI was 0.24–1.02 (p value = 0.046 using Fisher exact test). Among non-COVID patients, the OR was 0.64 and the exact 95% CI was 0.37–1.10 (p value = 0.103 using Fisher exact test).

The HR for COVID-19 patients was 0.55, the 95% CI was 0.31–0.98 (p value = 0.044). For non-COVID patients, the HR was 0.63 and the 95% CI was 0.40–1.00 (p value = 0.052). Fig 3 shows the one-year Kaplan–Meier estimated survival curves for COVID-19 and non-COVID patients: HAH versus their controls.

## Discussion

### Hospital-at-home as an alternative to in-hospital stay

The results of this study were reached after strict matching of the compared patients' groups. The scrutinizing match process has significantly diminished our study population reaching the final analysis and still, statistically significant results were reached, enabling further conclusions to be drawn relating to the study clinical outcomes. In face of our results, the initial motivation to show that HAH would be non-inferior to in-hospital stay, should be substituted with the notion that HAH, when delivered in the above-described meticulous methodology and appropriate technologies, could be superior to in-hospital stay in terms of less readmissions and longer patients' survival. It should be stated that during this study, we used technologies and staff that are being used in the routine HAH service in our medical center. All study results and the derived insights should be counted regarding the above. We foresee even better possible achievements as technology will continue to evolve and designated healthcare professionals gain even more experience in the realm of HAH.

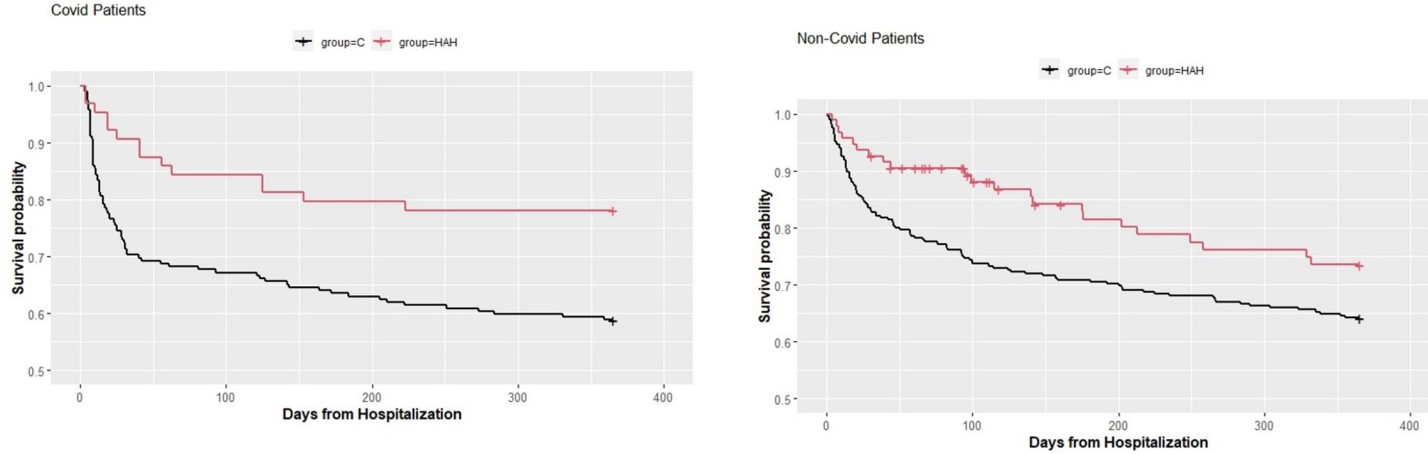

**Fig 3. Kaplan–Meier survival analysis.**

## The novelty of the described HAH setting

The main novelty in the above results is the fact that the HAH arm was based on telemedicine performed by the attending physician. Reliance on remote care technology enabled us to employ an experienced specialist in internal medicine, otherwise unavailable in case there was a need for large-scale home visits. We believe that the experience of the attending physicians was key to achieving superior clinical outcomes in the HAH arm over the C arm of this study. Also, the technologies employed were already used for several years (e.g. DATOS, a designated telemedicine-based platform) and already validated in clinical use cases. These technologies enable bridging the geographical gap between the physician and the patient. Moreover, the nursing staff, delivering both diagnostic and therapeutic measures to our patients is based on experienced nurses, almost all with several years of experience, holding advanced nursing degrees and qualifications.

## Clinical outcomes

Relating to the shortened length of stay in the HAH group: this could potentially help HAH organizations in their struggle to prove affordability. Nevertheless, we do not see the shorter LOS as a predominant achievement in the HAH bundle. As in-hospital departments become more crowded, length of in-hospital stay will inevitably become shorter–not due to better treatment but due to earlier, at times, too early patients' discharge. Therefore, we anticipate that HAH will prove to be as long as in-hospital stays and even longer. At home, the patients are not expected to make place to the next admissions and the length of hospitalization can and should stem only from the measures of good clinical practice.

Relating to the lower rates of re-admissions. These would have been easier to explain if indeed the HAH LOS were longer, providing the optimal treatment length needed. Since this was not the case, we assume that indeed, the experience of the attending physicians made the HAH hospitalizations more effective, putting the patients "on track" to better health, avoiding a larger number of re-admissions. It should be stated however that the retrospective nature of this study could be associated with a bias: it is possible that those patients that were suited to HAH continued to prefer their home environment and succeeded in maintaining themselves in the community while those who stayed in-hospital were more easily re-admitted. The fact that amongst COVID-19 patients there was no difference in readmissions could be related to

the fact that for many patients, in-hospital stay due to COVID-19 was compulsory by healthcare regulations and the same for re-admission. Therefore, the motivation of patients contributed less to this end point. This study was not designed to assess financial endpoints, typically affected by re-admissions. Such analyses should be sought in future, prospective studies.

Regarding the end point of survival at 30 days, HAH was associated with less mortality, in a statistically significant manner, for both COVID-19 and non-COVID patients. We assume that this difference stem from the higher chances of in-hospital acquisition of secondary infections, practically nonexistent in the HAH setting. However, it should be stated that hospital-aquiered infections and other, hospital–acquired complications were not monitored in this study nor we recorded the causes of death. Relating to the one-year survival rates: these continued to be significantly higher for HAH patients in the COVID-19 patients' group while losing statistical significance in the non-COVID-19 patients. This could be explained by the significant frailty characteristics of post-COVID-19 patients and the fact that on the one-year scale, in-hospital complications became less relevant for the non-COVID patients.

## The place of our results in perspective of current literature

Freund et al. (2023) compared early discharge of COVID-19 patients with controls and found out that a transference of such patients to their homes with continuing oxygen support was associated with shortened in-hospital stay but also with increased rate of readmissions and no benefits relating to long-term outcomes [26]. Their findings emphasize the difference between continuing medical attention and support in the community and the full bundle of services in the form of HAH. In their comprehensive meta-analysis, Chauhan, and McAlister (2022) reviewed 24 randomized clinical trials, including 10,876 patients, comparing post-discharge transference of patients to continuing attendance of virtual wards (VW) versus usual post discharge community care [27]. Although these were heterogenous services, none at the full scale HAH service, they found out that VW were associated with reduction in readmissions and lower healthcare costs. Nevertheless, favorable survival was shown only for patients suffering from congestive heart failure. Tierney et al. (2021) compared an acute care, home service for elderlies with continuing care within an elderly hospitalization unit [28]. In their 1-year analysis of 505 patients, they found out that the home care was associated with higher readmission rates and higher mortality at 30 days, 3- and 6-months duration. They concluded that their results stemmed from the fact that their home-care patients' population had a higher proportion of dependent, frail older patients. Their findings emphasize the need for thorough populations' matching as done in our study. Leong et al. (2021) identified ten systematic reviews comparing conventional hospitalization with two models of HAH: early support discharge (ESD) and admission avoidance (AA) [29]. ESD services were found to have comparable mortality and readmissions' rates as in-hospital stay but were associated with shorter hospitalizations. AA services showed a trend towards lower mortality, comparable or lower readmission rates. In summary, it can be concluded, from the existing literature, that medical services, at patients' homes, are heterogenous and as they become more similar to the HAH service, they are anticipated to provide better clinical outcomes.

## Sensitivity and generalizability of results

The results of our study provide promising evidence for the efficacy of tele-medicine controlled HAH care for acutely ill patients. The generalizability of these findings to other jurisdictions must account for several factors.

The first factor concerns healthcare infrastructure and technology. Regions or hospitals with well-developed telecommunication networks, electronic health record systems, and

advanced remote monitoring technologies are more likely to achieve positive outcomes. Furthermore, effective tele-medicine implementation requires skilled healthcare professionals who are proficient in using digital tools and remote monitoring technologies. Variations in training, availability of skilled personnel, and access to advanced technology across different regions could impact the consistency and effectiveness of HAH programs.

The second factor is cultural and socioeconomic considerations. Patient demographics, cultural attitudes towards tele-medicine, and socioeconomic status influence the acceptance and effectiveness of HAH programs. In the current study, that was a single-center study, patients in both groups came from the same area, the Dan district in central Israel. Therefore, both groups were similar in their demographics, including financial capabilities and social status.

The third factor includes the hospital location, which can affect the feasibility and efficiency of HAH care. In regions where patients are located far from healthcare facilities, providing timely tele-medicine services may be challenging, especially if patients deteriorate and need urgent immediate care. Areas with a higher density of healthcare resources, however, may see better integration and outcomes.

## Costs and resources analysis

As a retrospective analysis this study was not powered to produce a thorough cost analysis. Nevertheless, since all patients were treated by the same medical center, either in-hospital or by the HAH service, we can compare the resources needed for both treatment arms. While the net cost of one hospitalization day in-hospital is estimated at 2,830 NIS (~760 USD), the net cost of an HAH hospitalization day is only 1,660 NIS (~ 445USD) reflecting a 41.5% lowering of costs when moving from in-hospital to the HAH settings. Moreover, the HAH service personnel are engaged in other in-hospital activities and provide telemedicine services aside from the HAH service.

## Future prospects of advanced technologies and artificial intelligence in the tele-health sector

The telemedicine controlled HAH service described in this manuscript included application of several advanced, designated tele-health technologies: we used a tele-health dedicated platform for monitoring and documentation of vital signs and for video calls with our patients (DATOS). We also used TytoCare system as a digital recording stethoscope and a 6-lead electrocardiography machine for patients' self-usage. All of the above were previously validated by us in the HAH settings [10–14]. We did not use AI-based computing capabilities in this study. Several publications however reviewed the potential of assimilating AI into different tele-ehealth domains: tele-diagnosis, tele-interactions, and tele-monitoring [30, 31]. Widely described as the fourth industrial revolution, authors emphasize the potential obstacles facing whoever would assimilate AI into its telehealth services: including conflicts of patients' privacy, transparency, and safety concerns [32]. In their recent update on ethics and governance of artificial intelligence for health, the WHO (world health organization) declared that the following principles should guide the use of AI in health: autonomy protection, promotion of human well-being, human safety and the public interest, ensuring transparency, explainability and intelligibility, foster responsibility and accountability, ensuring inclusiveness and equity and promoting AI that is responsive and sustainable [33].

## Conclusions

This study offers compelling evidence supporting the effectiveness of home telemedicine-based hospitalization as a viable alternative to in-hospital internal medicine hospitalization.

The results indicate a significantly shorter LOS without significant difference in readmission rates for both COVID-19 and non-COVID patients in home-hospitalization. Both COVID-19 and non-COVID patients receiving home hospitalization showed a significant reduction in the risk of death. The results of this study support further research in the field, preferably in the setting of prospective, controlled randomized studies.

## Limitations

This study was conducted as a retrospective study at a single center. Consequently, even though we employed a thorough matching of patients' groups, it is imperative that our results and conclusions undergo further investigation due to probable bias of patient selection. The necessity for future prospective, randomized controlled trials, where patients are randomly assigned to either home or in-hospital hospitalization, is underscored to provide more robust evidence. Several limitations were also described earlier in the "sensitivity and generalizability of results" paragraph of the discussion.

## Author Contributions

**Conceptualization:** Noa Zychlinski, Yair Goldberg, Galia Barkai, Eyal Zimlichman, Gad Segal.

**Data curation:** Noa Zychlinski, Ronen Fluss, Daniel Zubli, Galia Barkai, Gad Segal.

**Formal analysis:** Noa Zychlinski, Ronen Fluss, Yair Goldberg, Gad Segal.

**Investigation:** Noa Zychlinski, Ronen Fluss, Daniel Zubli, Galia Barkai, Gad Segal.

**Methodology:** Noa Zychlinski, Ronen Fluss, Yair Goldberg, Daniel Zubli, Galia Barkai, Eyal Zimlichman, Gad Segal.

**Project administration:** Noa Zychlinski, Galia Barkai, Eyal Zimlichman.

**Software:** Yair Goldberg, Galia Barkai.

**Supervision:** Galia Barkai, Eyal Zimlichman.

**Validation:** Noa Zychlinski, Yair Goldberg, Eyal Zimlichman, Gad Segal.

**Writing – original draft:** Noa Zychlinski, Ronen Fluss, Yair Goldberg, Daniel Zubli, Galia Barkai, Gad Segal.

**Writing – review & editing:** Noa Zychlinski, Ronen Fluss, Yair Goldberg, Galia Barkai, Eyal Zimlichman, Gad Segal.

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
