## [Decision Letter · Decision Letter 0]

4 Jun 2024

PONE-D-24-14164Tele-medicine Controlled Hospital at Home is Associated with Better Outcomes than Hospital Stay. A Retrospective, Matched Study.PLOS ONE

Dear Dr. Segal,

Thank you for submitting your manuscript to PLOS ONE. After careful consideration, we feel that it has merit but does not fully meet PLOS ONE’s publication criteria as it currently stands. Therefore, we invite you to submit a revised version of the manuscript that addresses the points raised during the review process.

**ACADEMIC EDITOR: **The article is interesting but in order to be published it requires important modifications following the suggestions of the reviewers. Specifically, I believe that it is very important to carry out a statistical analysis and cost analysis. I also suggest evaluating the effect of Artificial Intelligence on telehealth. Moreover I suggest improving the bibliography. The decision is justified on PLOS ONE’s publication criteria. .

We look forward to receiving your revised manuscript.

Kind regards,

Filomena Pietrantonio

Academic Editor

PLOS ONE

Journal Requirements:

"The Research was supported in part by an Israel Science Foundation [Grant 277/21] and the Israel National Institute for Health Policy Research [Grant 2021/160/R].

Guarantor: No guarantees were given regarding this study."

5. We note that you have indicated that there are restrictions to data sharing for this study. For studies involving human research participant data or other sensitive data, we encourage authors to share de-identified or anonymized data. However, when data cannot be publicly shared for ethical reasons, we allow authors to make their data sets available upon request. For information on unacceptable data access restrictions, please see http://journals.plos.org/plosone/s/data-availability#loc-unacceptable-data-access-restrictions. 

**Additional Editor Comments:**

The article is interesting but in order to be published it requires important modifications following the suggestions of the reviewers. Specifically, I believe that it is very important to carry out a statistical analysis and cost analysis. I also suggest improving the bibliography.

Reviewers' comments:

Reviewer's Responses to Questions

**Comments to the Author**

1. Is the manuscript technically sound, and do the data support the conclusions?

Reviewer #1: Yes

Reviewer #2: Yes

2. Has the statistical analysis been performed appropriately and rigorously? 

Reviewer #1: No

Reviewer #2: Yes

3. Have the authors made all data underlying the findings in their manuscript fully available?

Reviewer #1: Yes

Reviewer #2: Yes

4. Is the manuscript presented in an intelligible fashion and written in standard English?

Reviewer #1: Yes

Reviewer #2: Yes

5. Review Comments to the Author

Reviewer #1: The article is interesting and well written. In order to imporve its quality I have some suggestions:

1. authors should highlight how results can chage depending on the type of technology, mode of telemedicine session administration and healthcare professional involved

2. authors should perform an inferential analysis wioth the use of regression techniques in order to investigate how outcomes can be associated to baseline charachteristics and socio econopmic indicators ( ie income, education, marital status...)

3. authors should perform scenario analysis or at least discuss on whether results are sensitive and generalizable to other jurisdications

4. authors should include a cost analysis highilithing how costs and in generalr esource use can change depending how the program is dimensioned (ie : initial investment rrequires fixed costs that should be compensated over different years, thus allowing to achieve a break even point )

Reviewer #2: Dear Authors, thank you for your work. please add more references (eg: 10.3390/s23125408, 10.3390/ijerph181910328, 10.1093/eurpub/ckab165.159), they should be >50. Did you consider AI effect on tele-health?

6. PLOS authors have the option to publish the peer review history of their article (what does this mean?). If published, this will include your full peer review and any attached files.

Reviewer #1: **Yes: **Matteo Ruggeri

Reviewer #2: No

---

## [Author Response · Author response to Decision Letter 0]

3 Jul 2024

Dear Editor and Reviewers, 

Your comments were all addressed and as a result, the manuscript is better!

On behalf of all authors, I thank you for considering our work for publication in PLOS ONE. 

Prof. Gad Segal, MD

---

## [Decision Letter · Decision Letter 1]

6 Aug 2024

Tele-medicine Controlled Hospital at Home is Associated with Better Outcomes than Hospital Stay. A Retrospective, Matched Study.

PONE-D-24-14164R1

Dear Dr. Gad Segal,

We’re pleased to inform you that your manuscript has been judged scientifically suitable for publication and will be formally accepted for publication once it meets all outstanding technical requirements.

Kind regards,

Filomena Pietrantonio

Academic Editor

PLOS ONE

Additional Editor Comments (optional):

All requests made by reviewers have been fulfilled and the article is eligible for publication.

Reviewers' comments:

Reviewer's Responses to Questions

**Comments to the Author**

1. If the authors have adequately addressed your comments raised in a previous round of review and you feel that this manuscript is now acceptable for publication, you may indicate that here to bypass the “Comments to the Author” section, enter your conflict of interest statement in the “Confidential to Editor” section, and submit your "Accept" recommendation.

Reviewer #1: All comments have been addressed

Reviewer #2: All comments have been addressed

2. Is the manuscript technically sound, and do the data support the conclusions?

Reviewer #1: Yes

Reviewer #2: Yes

3. Has the statistical analysis been performed appropriately and rigorously? 

Reviewer #1: Yes

Reviewer #2: Yes

4. Have the authors made all data underlying the findings in their manuscript fully available?

Reviewer #1: No

Reviewer #2: Yes

5. Is the manuscript presented in an intelligible fashion and written in standard English?

Reviewer #1: Yes

Reviewer #2: Yes

6. Review Comments to the Author

Reviewer #1: the manuscript is now suitable for publication. Authors have addressed all comments and recommendations properly

Reviewer #2: thank you to address each comment! your paper is better than the oldest version, and it is now ready to publish

7. PLOS authors have the option to publish the peer review history of their article (what does this mean?). If published, this will include your full peer review and any attached files.

Reviewer #1: **Yes: **matteo ruggeri

Reviewer #2: No

---

## [Editor Report · Acceptance letter]

9 Aug 2024

PONE-D-24-14164R1 

PLOS ONE

Dear Dr. Segal, 

I'm pleased to inform you that your manuscript has been deemed suitable for publication in PLOS ONE. Congratulations! Your manuscript is now being handed over to our production team.

Kind regards, 

on behalf of

Dr. Filomena Pietrantonio 

Academic Editor

PLOS ONE